# Nutrient Composition and Feed Hygiene of Alfalfa, Comparison of Feed Intake and Selected Metabolic Parameters in Horses Fed Alfalfa Haylage, Alfalfa Hay or Meadow Hay

**DOI:** 10.3390/ani14060889

**Published:** 2024-03-13

**Authors:** Marlene Köninger, Astrid von Velsen-Zerweck, Carolin Eiberger, Christof Löffler, Anja Töpper, Christian Visscher, Bernd Reckels, Ingrid Vervuert

**Affiliations:** 1Institute of Animal Nutrition, Nutrition Diseases and Dietetics, University of Leipzig, An den Tierkliniken 9, 04103 Leipzig, Germany; marlenekoeninger@googlemail.com; 2Main and State Stud Marbach, Gestütshof 1, 72532 Gomadingen-Marbach, Germany; poststelle@hul.bwl.de (A.v.V.-Z.); carolin.eiberger@hul.bwl.de (C.E.); 3Agricultural Center Baden-Württemberg of Cattle, Grassland Management, Dairy Farming, Fisheries and Wildlife (LAZBW), Atzenberger Weg 99, 88326 Aulendorf, Germany; christof.loeffler@lazbw.bwl.de; 4Agricultural Technology Center Augustenberg, Neßlerstraße 25, 76227 Karlsruhe, Germany; anja.toepper@ltz.bwl.de; 5Institute for Animal Nutrition, University of Veterinary Medicine Hannover, Bischofsholer Damm 15, 30173 Hannover, Germany; christian.visscher@tiho-hannover.de (C.V.); reckels@vets-wettringen.de (B.R.)

**Keywords:** feed intake, feed hygiene, short-chain fatty acids, crude nutrients, calcium

## Abstract

**Simple Summary:**

Increased periods of drought and the resulting shortage of forage necessitate the development of new strategies to find forage alternatives for equine nutrition. Alfalfa (*Medicago sativa*) is a heat-tolerant and high-yielding plant that provides high levels of protein and minerals, such as calcium (Ca) and magnesium (Mg). Alfalfa hay (AH) is an adequate forage alternative to meadow hay that seems to positively impact the fermentation profile in the large intestine in horses. However, little is known about feeding wrapped alfalfa as an alternative forage source in equine diets. Furthermore, its preservation in the form of haylage minimises leaf losses compared with that of alfalfa hay. The aim of this study was to examine the nutrient composition and feed hygiene of alfalfa as well as investigate the feed intake, blood, urine and faecal parameters of horses fed alfalfa haylage (AS) compared with AH and meadow hay (MH). Voluntary feed intake, nutrient values and faecal quality support AS as a suitable forage source in equids. In addition, the intake of wrapped forage provides a dust-free feed, which may be beneficial for horses suffering from equine asthma. Furthermore, its preservation in the form of haylage minimises leaf losses compared with that of alfalfa hay.

**Abstract:**

The aim of this study was to examine the nutrient composition and feed hygiene of alfalfa as well as investigate the feed intake, blood, urine and faecal parameters of horses fed alfalfa haylage (AS) compared with alfalfa hay (AH) and meadow hay (MH). A total of 11 geldings were fed ad libitum (2.1% dry matter (DM) of body weight (BW)) with alfalfa haylage, alfalfa hay and meadow hay (MH) in a Latin square design. On days 0 and 21 of the feeding period, blood samples were analysed for kidney and liver parameters. Faecal samples were analysed for pH, DM and short-chain fatty acids (SCFAs). Spontaneous urine was collected during the feeding period to analyse selected parameters. Forage was analysed in terms of feed hygiene and crude nutrients. In several feed samples of AS, AH and MH, the microbial reference ranges were exceeded for product-typical and spoilage-indicating bacteria and fungi. Crude nutrient analyses revealed a median crude protein content of 139 (138/142) g/kg DM for AS, which was similar to that in AH (127–135 g/kg DM) and substantially higher than in MH (79.1–87.7 g/kg DM). The calcium level in AS (11.3 g/kg DM) was significantly higher than that in MH (4.00–4.95 g/kg DM) but not compared with that in AH (9.80–10.4 g/kg DM). All blood parameters were within the reference ranges. Fractional excretion (FE)_Ca_ for AS-fed horses ranged from 8.13 to 22.0%; the FE_Ca_ for AH-fed horses ranged from 6.48 to 24.8%; the FE_Ca_ for MH-fed horses ranged from 6.69 to 53.2%. No significant differences were found in faecal pH or SCFA content in AS-fed horses compared with AH-fed and MH-fed horses. We concluded that alfalfa haylage provides an alternative forage for equine nutrition.

## 1. Introduction

Traditionally, meadow hay and pasture grass have been the major forage types used in equine feed rations [1]. To ensure the minimum daily amount of feed intake of, for example, hay, haylage, or silage, 1.5–2 kg dry matter (DM)/100 kg body weight (BW) is recommended, especially for stabled horses [2].

Climate change, in the form of a higher risk of droughts, has increased the need for forage alternatives and alternative forage production strategies to compensate for the lack of roughage for horses [3,4,5].

Alfalfa (*Medicago sativa*) is a heat-tolerant and high-yielding plant that provides high levels of protein and minerals, such as calcium (Ca) and magnesium (Mg). Alfalfa prefers calcareous soils with an alkaline pH. Alfalfa meets the nutrient needs of horses during high-energy-demand life stages, such as growth and lactation [6,7]. Alfalfa hay (AH) is an adequate forage alternative to meadow hay that seems to positively impact the fermentation profile in the large intestine in horses [8,9].

In different European countries, wrapped forage has partially replaced hay in equine diets [10,11,12,13]. Silage is a forage that is preserved under airtight conditions [14]. Under anaerobic conditions, lactate-producing microorganisms convert the sugar contained in the ensiled forage and reduce the pH, thus improving feed conservation [15]. The value of fermentation is linked to the dry matter content and water activity of the lactate-producing microbes [1,2]. According to Müller [5] haylage is defined as silage with a dry matter (DM) content of ≥ 500 g DM kg. Alfalfa is a plant material that does not easily compact and has a high buffer potential due to its high protein and Ca contents and partly low sugar levels, which makes it difficult to ensile [16].

Forage quality is mainly influenced by harvest techniques and storage conditions, as well as the fertiliser used and crop management practices implemented [5]. For alfalfa haylage (AS), as a new feed, raw nutrient and microbial analyses are crucial to ensure the high hygienic quality of wrapped forage for horses and to avoid inducing respiratory diseases through mould spores, intoxication by mycotoxins, or other illnesses in the horse [1].

Studies on voluntary feed intake and the preference of forage conserved as silage or haylage are scarce in equines, and their results are contradictory. A study reported low voluntary intake and a preference for silage in horses [1]. For horses with respiratory diseases, for example, equine asthma, feeding silage or haylage to reduce dust inhalation is recommended, among other feeding strategies [17].

The aim of this study was to examine the nutrient composition and feed hygiene of alfalfa haylage, as well as to investigate the feed intake and blood, urine and faecal parameters of horses fed AS in comparison to those of horses fed AH and meadow hay (MH). We hypothesised that voluntary feed intake and faecal quality support AS as a suitable forage source in equids.

## 2. Materials and Methods

### 2.1. Animal Welfare Statement

The project was approved by the Ethics Committee for Animal Rights Protection of the Tübingen District Government (TVA–Nr. Marbach 01/21 G) in accordance with German legislation for animal rights and welfare.

### 2.2. Study Design

From November 2021 until April 2022, this study was conducted to evaluate the feed intake, BW gain and faecal quality of horses fed AH, AS and MH at the Main and State Stud Marbach. 11 geldings were randomly fed AH, AS, or MH ad libitum in a three-week trial for each type of forage. A two-week wash-out period was performed between treatments by feeding MH at 1.5 kg dry matter (DM)/100 kg BW. The roughage was weighed daily (G&G PSB 150 kg/50 g, Kaarst, Germany) and offered three times per day in a separate trough to guarantee ad libitum intake and to limit leaf loss. After 24 h, the leftovers were reweighed and documented.

### 2.3. Animals/Husbandry

A total of 11 three-year-old clinically healthy warmblood geldings, with an average BW (± standard deviation) of 550 ± 38.7 kg at the start of the study, were kept in individual boxes and bedded on straw. The horses were exercised for 30 min daily according to a standardised riding protocol. Once per week, the horses had access to a winter pasture covered with snow for approximately 3 h.

### 2.4. Diet

Ad libitum intake was defined when leftovers were measured after 24 h of forage provision. Additionally, horses were fed 0.5 kg oats three times daily and 100 g of commercial mineral supplement (Josera Joker Mineral, foodforplanet GmbH & Co. KG^®^, Kleinheubach, Germany, composition see Appendix A). Horses had ad libitum access to water from an automatic water supplier.

### 2.5. Sampling

Blood samples were collected with a serum tube (S-Monovette, 9 mL, Sarstedt, Nümbrecht, Germany) on days 0 (d0) and 21 (d21) between 1 pm and 2 pm during each feeding trial from the left jugular vein (*V. jugularis externa sinistra)*. Spontaneously defecated fresh faecal samples (in total, *n* = 66 samples) were collected on d0 and d21 during each feeding trial. Urine samples from spontaneous urination were collected during each feeding trial by catching middle stream urine with a container. It was not possible to collect urine from all animals. Therefore, different numbers of urine samples (in total, *n* = 15 samples) were obtained from the horses in each forage group (see Table 11).

#### 2.5.1. Body Weight

BW was determined during each feeding trial on days 0 (d0), 7 (d7) and 14 (d14) using a mobile scale (TPW MOBIL1808 NE, T.E.L.L.-Steuerungssysteme GmbH&Co. KG^®^, Verden, Germany) and a display device (EAG80, T.E.L.L.-Steuerungssysteme GmbH&Co. KG^®^, Verden, Germany). Unfortunately, it was not possible to measure BW on day 21, as the scale was not available on the stud farm.

#### 2.5.2. Forage

The different types of forages were cultivated in the same field as the Main and State Stud Marbach. All forage was harvested in summer 2021, and each forage type was maintained within one batch. After harvesting, AH was artificially dehydrated (Henkel GBR, Neufra, Germany). For haylage production, the fresh alfalfa plants were cut to a cutting height of at least 10 cm. The cropped plants were turned once and swathed. In a baler–wrapper combination (Göweil G-1 F125 Kombi, Göweil Maschinenbau GmbH, Kirchschlag, Austria), the alfalfa was pressed into round bales with a chop length of 35 mm with the addition of a silage additive (Farmacid-NCP 5, Konsil Europe, Wettin-Löbejün, Germany). The additives included propionic acid, sodium propionate and sodium benzoate. A bottom film was used first, and subsequently, a wrapping film was wrapped in at least 10 layers. Samples of AH (*n* = 3), AS (*n* = 9) and MH (*n* = 3) for nutrient and hygienic analyses were collected according to VDLUFA guidelines [18] using a drill (H-Dry Twister, Moisture measurement Dietmar Hipper weigh-dose-analyse e.K., Bad Saulgau, Germany).

### 2.6. Analysis

#### 2.6.1. Blood Analysis

Blood samples were allowed to clot at room temperature for at least 1 h and centrifuged at 2500× rpm for 10 min (EBA 3 S Hettich, Kirchenlengern, Germany). Serum was transferred to 1.5 mL tubes (Eppendorf AG^®^, Hamburg, Germany) and stored at −18 °C until analysis.

Serum parameters (total protein, albumin (ALB), aspartate aminotransferase (AST), bilirubin, creatinine (Crea), gamma-glutamyl transferase (GGT), glutamate dehydrogenase (GLDH), urea, potassium (K), sodium (Na), phosphorus (P), calcium (Ca) and chloride (Cl)) were analysed using an automated chemistry analyser (Roche Cobas C311, Roche Diagnostic GmbH, Mannheim, Germany). Bile acids were analysed with an enzymatic colour test (Laboratory and Technology Eberhard Lehmann GmbH, Berlin, Germany).

#### 2.6.2. Urine Analysis

Urine samples from spontaneous urination were collected throughout the feeding trial, transferred into 15 mL tubes and frozen at −18 °C until analysis. Urine parameters (Ca, P, K, Cl, Na and creatinine (Crea)) were analysed using a chemistry analyser (Roche Cobas C311, Roche Diagnostic GmbH, Mannheim, Germany). For the analysis of Ca and P, urine samples were acidified with 1 mL of HCL (dilution 1:1).

To calculate the fractional excretion of electrolytes (Ca, P, Cl, K and Na), the following formula was used [19]: (1)$$Fractional\;excretion\left(FE\right)\;\%=\frac{Electrolyte\;Urine}{Electrolyte\;Serum}\;\times \;\frac{Creatinine\;Serum}{Creatinine\;Urine}\;\times \;100$$

To calculate fractional excretion (FE) of the electrolytes, blood parameters from d0 were used to calculate FE from days 0–10 and blood parameters from d21 were used for calculating excretion from days 11 to 21.

#### 2.6.3. Faecal Sensorics 

Faecal samples were sensorially examined according to the faecal score (Table 1). 

#### 2.6.4. Faecal pH

Immediately after the sensory examination of the faeces, free faecal liquid was squeezed out of the faeces for the measurement of pH. The pH in the squeezed-out faecal water was determined with a digital pH device (pH CHECK, Dostmann electronic GmbH^®^, Wertheim-Reicholzheim, Germany) in triplicate measurements. The subsample from which the liquid was pressed off was subsequently discarded. The remaining original faeces were stored at −18 °C until analysis.

#### 2.6.5. Faecal DM

We weighed 10 g of faeces (PG5002 Delta Range, Mettler Toledo^®^, Columbus, OH, USA), which were then dried for at least 12 h at 105 °C in a circulating air-drying cabinet (UFE 700, Memmert GmbH^®^, Schwabach, Germany) in triplicate and reweighed.

#### 2.6.6. Faecal Short-Chain Fatty Acids (SCFAs)

A total of 20 mg of faeces and 20 mL of *Aqua dest*. were mixed in a beaker (mixing ratio 1:1). Faecal water was squeezed out, transferred into a 10 mL tube, and centrifuged at 4000 rpm for 10 min (EBA 3 S Hettich, Kirchenlengern, Germany). The filtered supernatant (1 mL) was mixed with 100 μL of an internal standard (initial solution: 10 mL of 17% phosphoric acid and 0.025 mL of 4-methylvaleric acid) and frozen at −18 °C until analysis. The short-chain fatty acids (SCFAs) acetate (C2), propionate (C3), isobutyrate (IC4), butyrate (C4), isovalerate (IC5), valerate (C5) and caproate (C6) were analysed via gas chromatography (GC-2014, Shimadzu, Duisburg, Germany). The sample was separated along a 30 m long column (Restek, Stabilwax-DA, Capillary GC Column) in a gas chromatograph (GC-2014, Shimadzu, Duisburg, Germany). The column temperature was 225 °C (injector temperature was 220 °C, detector temperature was 240 °C). SCFAs were analysed in duplicate using a flame ionisation detector over a 12 min analysis period in the order of acetate, propionate, isobutyrate, butyrate, isovalerate, valerate and caproate with reference to the internal standard.

#### 2.6.7. Crude Nutrients in Feedstuffs

Crude nutrients in feedstuffs were assayed using the Weende system [15]. DM was determined after oven-drying (103 °C). The contents of neutral detergent fibre after amylase treatment and ashing (aNDFom), acid detergent fibre after ashing (ADFom) and acid detergent lignin (ADL) were determined according to VDLUFA [18]. The content of nitrogen-free extractives (NFE) was calculated as
NFE = DM − (CA + CP + CL + CF)(2)
where CA is crude ash, CP is crude protein, CL is crude lipid and CF is crude fibre.

Ca, P, K and Mg contents were analysed according to VDLUFA [18]. Metabolizable energy (ME) was calculated as (ME (MJ/kg DM) = −3.54 + 0.0129 CP + 0.042 CL − 0.0019 CF + 0.0185 NFE) [6]. 

#### 2.6.8. Microbiological Examination of Feedstuffs

Feed samples were sensorially examined using the standard VDLUFA method (SOP-LTZ). According to the VDLUFA method book (MB) III 28.1.2 (2012), bacteria and fungi were examined according to microbial group (MG 1–7) (Table 2). To determine feed quality and hygiene, samples were classified according to the quality level in the VDLUFA guidelines [20,21].

#### 2.6.9. Fermentation Parameters in AS

The fermentation parameters of AS were analysed according to VDLUFA MB III 18.1–18.5 (2012) [20].

In the form of a score system, the ensiling quality was assessed based on the sensory and chemical analysis of the forage [22]. The butyrate, acetate, propionate, pH (according to the DM content) and NH3-N content parameters were used for assessment.

### 2.7. Statistics

SPSS^®^ (Version 27, IBM, Armonk, NY, USA) and STATISTICA^®^ (Version 14, TIBCO Software, Palo Alto, CA, USA) were used to analyse the data. All data were tested for normal distribution using the Shapiro–Wilk test. Post hoc power analysis for the main effects, such as faecal DM, revealed a minimum sample size of 9 horses. ANOVA with covariance and a post hoc test (Fisher’s LSD) were conducted for normally distributed data to determine significant differences between horses fed AH, AS and MH on days 0 and 21 of the trial. For non-normally distributed data (pH and faecal SCFAs), the Kruskal-Wallis test with Bonferroni correction was performed to determine significant differences between the AH, AS and MH treatments on days 0 and 21. For crude nutrients, feed intake and urine parameters, a univariate ANOVA with a post hoc test (Fisher’s LSD) was performed for normally distributed data. For non-normally distributed data, a Kruskal-Wallis test with Bonferroni correction was conducted to identify differences between the AH, AS and MH treatments.

*p* < 0.05 was considered statistically significant.

Normally distributed data are presented as the mean ± standard deviation (SD). Non-normally distributed data are expressed as the median and [25th/75th percentiles].

## 3. Results

### 3.1. Crude Nutrients

The DM was significantly higher in AH and MH than in AS (Table 3). The CP level did not differ between AH and AS but was significantly lower in MH. The Ca level was significantly higher in AS than in MH. AH and MH showed no significant differences in the Ca level. The P, K and Mg contents did not significantly differ among the three forage types (Table 3). Energy levels showed no significant differences among the forage types (Table 3).

According to the Weissbach and Honig scoring system [22], AS received three out of five points, which indicates average quality. The deduction of points was mainly due to the high DM content (69.2%) and high pH (5.96) in the AS (Table 4). 

### 3.2. Microbial Hygiene

For one sample of AH and one sample of MH, the reference limit of ≤3.0 × 10^7^ CFU/g was exceeded for microbial group (MG) 1. For two samples of AH and one sample of MH, the benchmark for MG 4 was exceeded. The values for all AH samples were in the reference ranges for MG 2, 3 and 5 (Table 5). The values for all samples of MH were in the reference ranges of MG 2 and 3. One sample of MH was above the benchmark for MG 5, and all MH and AH samples exceeded the reference ranges for MG 7 (Table 5).

The reference ranges for MG 1 and MG 2 were exceeded in four of the AS samples. Five AS samples were above the reference limit of 0.01 × 10^6^ CFU/g for MG 3. One AS sample exceeded the reference ranges for MG 4, 5 and 6. In total, two AS samples were above the reference range for MG 7, which is a maximum of 2 × 10^3^ CFU/g of yeast. None of the samples showed an increased level of clostridia. The limit of lactic acid bacteria was <1.0 × 10^4^ to 4.6 × 10^7^ per gramme of feed (Table 6).

### 3.3. Feed Intake

Daily feed intake varied from 2.10% (AH) to 2.15% (AS) of BW and was not significantly different among the forages (Table 7). 

### 3.4. Body Weight

The horses showed no significant difference in BW gain when fed AH, AS, or MH during feeding periods (Table 8).

### 3.5. Blood Parameters 

All blood parameters except Ca were within the reference ranges at both time points. Ca levels exceeded the reference ranges on d21 for AH and AS and for MH on d0 (Table 9 and Table 10).

### 3.6. Urine Parameters

Only urine samples from spontaneous urination were obtained from the horses; therefore, different numbers of urine samples were obtained from the horses in each forage group. As shown in Table 9, serum electrolyte levels and creatinine levels were similar throughout all time points (d0 and d21) in all horses. The calculated fractional excretion of electrolytes when feeding AH, AS and MH is shown in Table 11.

### 3.7. Faecal DM

On d0, no difference was found in the faecal DM among the diets. On d21, horses fed AH (21.7 ± 1.48%) and AS (21.2 ± 2.09%) showed a significantly higher faecal DM level than those fed MH (19.5 ± 1.71%) (Figure 1).

### 3.8. Faecal pH

On d0, no difference was found in the faecal pH among the diets. Horses fed AH had a significantly higher median faecal pH on d21 (7.47 (7.17/7.67) than horses fed MH (6.90 (6.57/7.17)) (*p* = 0.004). Horses fed AS (7.23 (6.90/7.50)) showed no significant difference in their median faecal pH compared with those fed MH on day 21, but a tendency (*p* = 0.151) was observed. No significant difference was found in faecal pH between horses fed AH and AS on d21 (Figure 2).

### 3.9. Faecal SCFA

For acetate, propionate, butyrate (i- and n-), and valerate (i- and n-), the faecal SCFAs did not significantly differ among the horses on d0. Horses fed MH had a higher propionate level on d21 than horses fed AH. Horses fed AS had similar SCFA levels to those fed AH or MH on d21 (Table 12).

## 4. Discussion

In this study, the nutrient composition and microbial hygiene of AH, AS and MH, as well as the effects of their feeding on blood, urine and faecal parameters with ad libitum access, were investigated in exercising horses.

Bale-wrapped forage sealed in airtight bags for anaerobic fermentation is an appropriate forage for horses in Europe [2], as well as in other non-European countries [26]. Alfalfa is a high-quality forage, and its preservation in the form of silage or haylage minimises leaf loss compared with that of hay [27]. Furthermore, as a low-dust forage, AS contributes to the health of horses with respiratory diseases, especially those with equine asthma [17].

Due to the high buffer capacity of the protein and Ca in alfalfa, as well as its low sugar content, alfalfa is a weakly fermentable plant [9,27]. In our study, attention was paid to a cutting height of at least 10 cm to avoid contamination by dirt in the forage. Additionally, cutting too deeply damages the above-ground rootstock, reduces resprouting, and thus the stand density. To prevent leaf loss, gentle handling when turning the alfalfa and forming a swath after cutting were applied.

With the use of a press-wrapping combination and cutting the alfalfa stems to 35 mm, increased compaction of the coarse structure of the plant material was achieved. Quality-tested films in the form of a bottom and a top film in at least 10-fold wrapping were used in alfalfa haylage production to avoid punctures of the plant stalks. If the surrounding stretch film layers do not provide an airtight seal, oxygen enters, resulting in fungal growth [1]. However, the use of plastic silage film is a disadvantage, as no sustainable materials are currently available.

The pressed bales were gently loaded, using bale grippers to prevent film damage. Optimally, alfalfa silage or haylage bales should be stored on a concrete surface in combination with a bird protection net, as we saw in our study. Furthermore, periodic bale inspection was conducted, which is recommended so that film integrity issues can be promptly addressed [28]. In our study, a fermentation additive (propionic acid) was applied, as other authors recommended to improve the silage process [29]. 

Wrapped forage with a lower DM content (30.9% and 57.7% DM) is at higher risk of bacterial contamination by, for example, yeast and clostridia, compared with dry forage such as hay [1,30]. However, Müller and Udén [30] showed that hay (88.4% DM) had a higher amount of Enterobacteria (260 × 10^3^ CFU/g FM) than haylage at 30.9% DM (not detected).

The dry matter content of haylage seems to be decisive for ensiling success. Haylage/silage bales with a dry matter content ≤55% had fewer sensory deficiencies obtained from regular routine sampling from the stud farm (personal communication). These observations confirm the findings of O’Brien et al. [31], who showed that a higher DM content was positively correlated with the presence of mould in baled silage. 

Jaster and Moore [32] reported higher contents of fermentation products (lactate 0.51% and acetate 0.02%) in alfalfa haylage (55% DM) stored in big silos than those found in this study. However, the median DM content of alfalfa haylage (69.2%) was higher in our study, which could be the reason for the lower fermentation activity than that reported by Jaster and Moore [32].

Müller et al. [33] reported a lactic acid level of 4.8 g/kg DM and an acetic acid level of 1.6 g/kg DM in a haylage with a DM content of 67%. The pH of the AS in our study (5.96) was similar to that in haylage, as reported by Müller et al. [33] (5.58). The ethanol content was higher (6.7 g/kg DM) in haylage [33] than in the AS in our study (1.07 g/kg DM). 

However, according to Müller et al. [33], pH and fermentation products are less reliable indicators of hygienic quality. The assessment of the hygienic quality of haylage is therefore more dependent on the number of microbes in the forage, but reference ranges for different microbial species are lacking, and their associations with health effects in horses are poorly understood [1]. Currently, no microbial reference ranges have been established for baled silage or haylage for horses made from 100% legumes, such as alfalfa. To classify the microbiological quality of the AS, we compared the microbial results with the reference ranges established for grass silage.

The samples of AS, AH and MH from our study showed elevated microbial levels (Table 5 and Table 6). To ensure the hygienic quality of forage, the regular microbial monitoring of feedstuffs, not just alfalfa products, is also required, as is the critical examination of the hazard points in production and storage. Several authors [34,35] have emphasised the need to focus on the hygienic quality of forage, especially for forage crops harvested late in season, because hygienic risks may increase owing to increased amounts of enterobacteria, clostridia, moulds and yeasts.

Alfalfa provides high levels of protein and minerals [36] and is well-established in the feeding of ruminants and horses [37].

The nutrient content in alfalfa widely varies depending on factors such as cultivation, harvesting and processing methods [38,39,40]. Some authors have reported a CP content in alfalfa hay of between 20.7% and 23.0% of the DM [38,41]. In other studies, CP levels ranged from 10.0% to 13.4% of the DM [8,39]. In the present study, the median CP contents of AH (13.2%) and AS (13.9%) were similar and did not align with the results of American studies [42]. For the formulation of an appropriate diet for horses, regional data on nutrient composition that consider various factors, such as soil, harvest and fertiliser management, are important [43] 

The dry matter intake capacity of horses varies between 2.0 and 3.8% of BW [15], and alfalfa is palatable to horses [41]. The results of our study show that despite the reported high palatability of alfalfa, horses did not show an increased intake of AH or AS compared with MH when offered feed ad libitum (Table 7). In contrast to our study, La Casha et al. [44] found that alfalfa hay intake was approximately 3.1% of the BW of yearlings and was significantly higher compared to the intake of Matua bromegrass hay (2.8% of BW) and coastal Bermuda grass hay (2.1% of BW). Similar to our study, Dulphy et al. [45] reported a voluntary daily intake of alfalfa hay (17.7% crude protein) of approximately 2.35% of the BW in adult geldings. The alfalfa hay feeding of 1.5 kg DM/100 kg BW showed that adult horses consume alfalfa hay faster than meadow hay [8]. Müller and Udén [30] reported that the rate of the consumption of silage (30% DM) was the highest and its feed intake period was longer compared with those of haylage (57.7–68.4% DM) and hay (88.4% DM), which had the lowest consumption rate. Our study showed a daily ad libitum feed intake of 2.1% of AS, which is low. Notably, 1.5 kg of oats (corresponding to 1.3 kg DM) was added to the daily ration. In addition, horses were bedded on straw, and the straw intake probably contributed to the dry matter intake. As the study was performed on a stud farm, an exchange of the bedding material, such as rubber mats, was not possible.

To date, no clear predictor in terms of the chemical components of hay for voluntary feed intake by horses for different types of forage has been detected [41,46]. Ralston [47] stated that factors such as stimulation by the smell and texture of feed or the time of day, as well as social structures, contribute to meal frequency and the duration of feed consumption by horses, which may be more important for long-term feed intake.

The recommended daily Ca intake for a 500 kg horse is ~17–18 g [6]. The median Ca content was the highest in AS (11.3 g/kg DM). The Ca level did not differ significantly from that in AH (10.2 g/kg DM) but was significantly lower than that in MH. However, the Ca content in alfalfa was lower than that reported in other studies (a Ca content of 26.3 g/kg DM) [48].

In our study, the daily Ca intake of horses with a median intake of 2.15% DM AS of their BW was approximately 122 g/day. Therefore, the Ca intake exceeded the daily recommendations in these horses by a factor of 6.98.

Even though the horses were oversupplied in this study, calcium blood levels were within the reference ranges, as calcium levels in the blood are subject to strict hormonal control by parathyroid hormone and calcitonin. Our data support the findings of Schryver et al. [49] that plasma calcium (and phosphorus) levels in horses are unaffected by variations in calcium intake due to strict hormonal control.

Compared with other species, horses excrete a much higher percentage of calcium through the kidney; thus, the kidney is an important regulatory organ in equine calcium metabolism [50].

The renal excretion of Ca is directly related to the amount of absorbed calcium [51]. In a study by Bickhardt et al. [52] with healthy horses, the reference FE_Ca_ ranged from 1.3 to 33.2%, which seems to be physiological for horses fed hay, oats and straw. In our study, horses fed alfalfa had an FE_Ca_ within the reference range given by Bickhardt et al. [52] but slightly higher than the one given by Morris et al. [24]. However, feeding MH was related to a higher FE_Ca_ (maximum: 53.2%), although Ca intake was lower than when feeding AS or AH (Table 11). The discrepancies in these results are not fully understood. Although the middle stream of urine was attempted to be taken, there are still uncertainties in the sample routine. As the high Ca intake by alfalfa might be a limitation, long-term studies are required to better define the maximum daily intake of alfalfa, either as hay, haylage or silage.

The recommended P intake for a 500 kg horse is 12.0 g/day [6]. The median P content in our study was highest in the MH treatment (3.75 g/kg DM) but did not differ significantly from that in the AS (3.30 g/kg DM) or AH (3.50 g/kg DM) treatments. Köninger et al. [8] reported a similar P content for alfalfa hay (3.80 g/kg DM) but a lower P content for MH (2.55 g/kg DM). Other authors [48] reported a lower P content of only 1.8 g/kg DM in alfalfa.

With a P intake of 35.5 g/day through ad libitum AS intake (2.15% DM of BW), the P intake of our horses exceeded the recommendation by a factor of 2.87. In our study, serum P levels were within the reference ranges for all diets (Table 9). Lumsden et al. [53] postulated a reference range for serum P in standardbred horses (with horses of different ages and sexes exposed to light and heavy training) of 1.03 ± 0.16 mmol/L, which is similar to that used in our study. Caple et al. [54] found no significant changes in serum Ca and P concentrations in horses fed diets containing different levels of Ca and P over a 120-day period. Schryver et al. [49] showed that only a very high P content in the ration (1.19%) significantly raised the P concentration in the plasma. Rations with lower phosphorus concentrations did not seem to impact the phosphorus plasma levels in those horses].

Schryver et al. [49] hypothesised that in horses and ponies, even wide variations in calcium intake have relatively little effect on P metabolism if the dietary level of P is adequate. 

In our study, the P concentration in the urine of horses fed MH ranged from 0.02 to 0.06 mmol/L and from 0.00 to 0.24 mmol/L for horses fed AS. The FE_P_ for horses fed AS in our study ranged from 0.00 to 0.02%; the FE_P_ for horses fed MH ranged from 0.00 to 0.01% (Table 11).

A calculated FE_P_ of 5.53 ± 0.97 in mares fed a diet of 35.1 g of Ca and 38.7 g of P per day (the diet contained 3 kg of oaten chaff, 3 kg of oats, 3 kg of bran and 0.18 kg of CaCO_3_) was described by Caple et al. [54]. In addition, an increased urine P concentration and calculated FE_P_ were observed by increasing the P intake from 38.7 g to 41.4 g P/day.

Caple et al. [54] mentioned that the Ca and P concentrations in the urine samples collected from groups of horses fed the same diet considerably varied. The calculated FE, however, eliminates the effects of any variations in the concentrations of minerals in urine; therefore, a reliable indication of mineral excretion can be obtained from a single urine sample [54]. In contrast, Lefebvre et al. [55] stated that FE measurements are rarely used in veterinary practice due to the large inter- and intraindividual variability in the FE, which considerably limits the usefulness of its measurement for the identification of tubule function abnormalities. Lefebvre et al. [55] attributed the high intraindividual variability in FE results to the excretion of any electrolyte being adapted to body requirements to maintain its plasma concentration within the normal range.

No adverse effects on feed intake or faecal quality of feeding AS were observed in the present study. On the contrary, the quality of faeces (pH and DM) improved compared with those obtained with the feeding of MH (Figure 1 and Figure 2). This result is also noteworthy in view of the fact that the horses were abruptly switched from hay to AS.

Müller [56] reported that horses fed grass silage had a significantly higher faecal pH than horses fed grass hay. Alfalfa seems to impact the fibre fermentation in the large intestine of horses [8]. Horses fed alfalfa hay had significantly higher faecal DM and pH than horses fed MH after 28 days of feeding [8]. In our study, the faecal DM content was significantly (*p* < 0.05) higher for horses fed AS than for those fed MH (Figure 2). The finding of a higher faecal DM by feeding AS might be of clinical relevance. For example, the so-called free faecal water syndrome (FFWS), where the horse exhibits 2-phase faeces expulsion, with an initial solid phase followed by a liquid phase, is strongly affected by forage quality [57]. As faecal DM was significantly higher in horses fed AS than feeding MH, alfalfa may have some potential in the treatment of FFWS in horses. This aspect deserves further attention in clinical studies.

When fed AS, horses did not have a higher median faecal pH (7.23) than horses fed MH (6.90) (*p* = 0.151). 

Muhonen et al. [58] found no difference in the colon or faecal pH of horses fed silage (36% DM) after an abrupt change from hay. 

The pH in the caecum content was lower when feeding AH than when feeding bromegrass hay to mature quarter horses [9]. Conversely, horses consuming alfalfa had a higher faecal pH than horses consuming bromegrass [9].

Muhonen et al. [59] reported that even if the total water intake (drinking + water in feed) of horses was higher on a silage diet than on a hay diet, the total water output per day in the faeces did not differ between the diets. However, we did not measure water intake, but this should be considered in further investigations. 

Horses use the products of enzymatic digestion, such as starch, in the small intestine as well as bacterial fermentation products, such as SCFAs, in the large intestine as sources of metabolised energy [47]. Evidence shows that changes in dietary patterns, especially adding alfalfa to the ration, alter the gastric and hindgut microbiota, subsequently leading to changes in the hindgut pH and fermentation patterns [60,61]. In a former study [8], horses fed AH had significantly higher faecal SCFA levels on day 28 than horses fed MH. These findings could not be confirmed for AS in the present study, as the faecal SCFA levels did not significantly differ among the horses fed different diets. 

Muhonen et al. [58] found that the SCFA concentrations in the colon between horses fed hay (81% DM), haylage (55% DM) and silage (36% DM) did not differ during a 28-day observation period in four adult colon-fistulated horses. The acetic acid, propionic acid, butyric acid and ethanol contents and pH of the haylage were similar to those of the AS in our study [58]. Miyaji et al. [62], who studied fibre digestibility and fermentation variables in different segments of the equine hindgut, reported that horses fed hay or silage from timothy sward showed no differences in the SCFA concentrations. In the hindgut segments, the apparent digestibility of DM and organic matter and fibre (neutral (NDF) and acid (ADF) detergent fibre fractions were similar between the diets [62].

Sorensen et al. [9] found elevated (*p* < 0.05) acetate, butyrate and total SCFA levels in the caecum of horses fed alfalfa hay ad libitum compared with horses fed bromegrass hay. The acetate concentrations in alfalfa-fed horses were higher in caecal than in faecal samples at several sampling times [9]. The concentrations of caecal acetate were higher in horses fed alfalfa hay than in horses fed bromegrass hay, even if the faecal acetate levels differed between the diets only at one sampling time point [9]. Silage, haylage and hay produced from the same grass crop tend to produce similar responses in microbial and chemical composition in the right ventral colon [63]. However, the faecal samples in the present study provided only limited insight into the digestive parameters; therefore, studies on AS diets should be conducted with fistulated horses.

## 5. Conclusions

Alfalfa haylage provides an alternative forage for equine nutrition in the context of climate change. Voluntary feed intake and nutrient values such as protein and faecal quality support alfalfa haylage as a suitable forage source in equids. In addition, the intake of wrapped forage provides a dust-free feed, which may be beneficial for horses suffering from equine asthma. Furthermore, its preservation in the form of haylage minimises leaf loss compared with that of alfalfa hay. Special attention should be paid to the harvesting and storage techniques, as alfalfa is difficult to conserve. In addition, alfalfa silage or haylage should be subject to regular hygienic and nutrient monitoring. 

It is also important that new findings on feedstuffs that have not yet been used in horses be disseminated to a wider audience, e.g., through digital communication channels.

## Figures and Tables

**Figure 1 animals-14-00889-f001:**
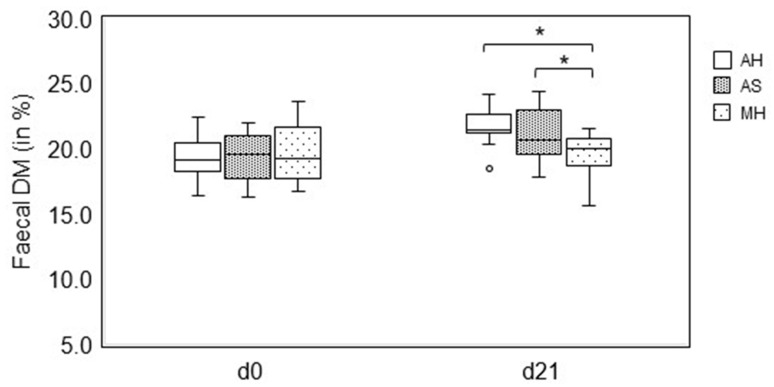
Comparison of the faecal dry matter (DM) among feeding alfalfa hay (AH), alfalfa haylage (AS) and meadow hay (MH) on day 0 (d0) and day 21 (d21). The box represents the first and third quartiles, the horizontal line represents the median; the upper and lower whiskers are a maximum of 1.5 times the interquartile range. Points (°) represent outliers. The symbols (*) above the brackets indicate significant differences (*p* < 0.05). Data were analysed by ANOVA with covariance and a post hoc test (Fisher’s LSD).

**Figure 2 animals-14-00889-f002:**
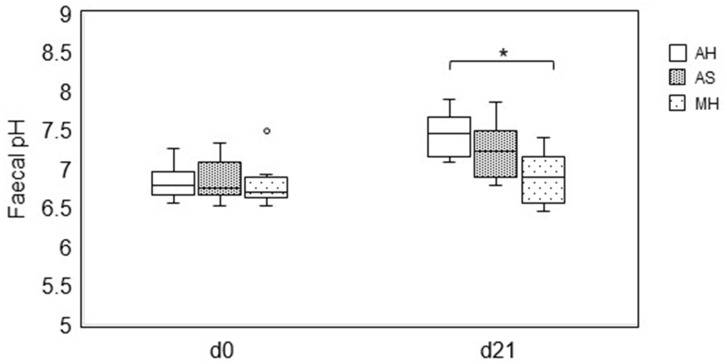
Comparison of the faecal pH among feeding alfalfa hay (AH), alfalfa haylage (AS) and meadow hay (MH) at d0 and d21. The box represents the first and third quartiles, the horizontal line represents the median; the upper and lower whiskers are a maximum of 1.5 times the interquartile range. Points (°) represent outliers. The symbols (*) above the brackets indicate significant differences (*p* < 0.05). Data were analysed by Kruskal-Wallis test with Bonferroni correction.

**Table 1 animals-14-00889-t001:** Faecal score of the sensory inspection of faeces.

Description	Score
Firm, easily definable balls of faeces, smooth, greenish/brownish colour, aromatic smell	0
Soft, pulpy balls of faeces, but still formable, greenish/brownish colour, aromatic smell	1
Mushy, loss of stool ball structure, greenish/brownish colour, aromatic smell	2
Mushy to watery, light green/brown colour, nonphysiological smell	3
Highly liquid, light green/brown colour, nonphysiological smell	4

**Table 2 animals-14-00889-t002:** Classification of bacteria and fungi during microbiological examination according to VDLUFA MB III 28.1.2 (2012) [20].

Group	Significance	Indicator Species
MG 1 ^1^	Product-typical	Yellow sprouts, *Pseudomonas/Enterobacteriaceae*, other product-typical bacteria
MG 2	Spoilage-indicating	*Bacillus*, *Staphylococcus/Micrococcus*
MG 3		*Streptomycetes*
MG 4	Product-typical	Blackness-fungi: *Acremonium*, *Fusarium*, *Verticillium*, *Aureobasidium* and other product-typical fungi
MG 5	Spoilage-indicating	Moulds: *Aspergillus*, *Penicillium*, *Scopulariopsis*, *Wallemia* and other spoilage-indicating fungi
MG 6		*Mucorales*
MG 7		Yeast (all types)

^1^ MG = microbial group.

**Table 3 animals-14-00889-t003:** Nutrient composition of the forage: Alfalfa hay (AH) and meadow hay (MH) expressed as single values; alfalfa haylage (AS) expressed as the mean ± SD.

Nutrients (g/kg DM ^1^)	AH (*n* = 3)	AS (*n* = 9)	MH (*n* = 3)
DM	942/935/944 ^a^	670 ± 60 ^b^	903/884/905 ^a^
CP ^2^	132/127/135 ^a^	139 ± 48 ^a^	82.0/79.1/87.7 ^b^
aNDF_OM_ ^3^	554/553/555 ^ab^	538 ± 12 ^a^	609/608/622 ^b^
Ca	10.2/9.80/10.4 ^a^	11.0 ± 1.4 [10.1/11.9] ^a^	4.10/4.00/4.95 ^b^
P	3.40/3.35/3.50 ^a^	3.30 ± 0.3 ^a^	3.60/3.45/3.75 ^a^
K	24.0/24.0/24.4 ^a^	23.0 ± 1.4 ^a^	25.0/22.5/26.8 ^a^
Mg	1.21/1.21/1.30 ^a^	1.40 ± 0.2 ^a^	1.00/1.00/1.30 ^a^
ME ^4^, in MJ ^5^	5.55/5.47/5.60 ^a^	5.57 ± 0.12 [5.42/5.66] ^a^	6.98/6.80/7.16 ^a^

^1^ DM = dry matter, ^2^ CP = crude protein, ^3^ aNDF_om_ = amylase treated neutral detergent fiber in organic mass, ^4^ ME = metabolizable energy, ^5^ MJ = Mega Joule. ^ab^ Different superscript letters indicate significant differences within a row (*p* < 0.05).

**Table 4 animals-14-00889-t004:** Fermentation parameters of alfalfa haylage (AS), expressed as the mean ± SD.

Parameter	AS (*n* = 9)
pH	5.96 ± 0.06
NH_3_-N, in g/kg DM	7.21 ± 0.62
Saccharose, in g/kg DM	3.99 ± 0.99
Glucose, in g/kg DM	15.9 ± 4.89
Fructose, in g/kg DM	14.9 ± 3.95
Total sugar, in g/kg DM	34.8 ± 8.74
Lactate, in g/kg DM	1.00 ± 1.13
Formic acid, in g/kg DM	<LOD ^1^
Acetate, in g/kg DM	1.28 ± 0.50
Propionate, in g/kg DM	0.82 ± 0.71
Butyrate, in g/kg DM	<LOD ^1^
Ethanol, in g/kg DM	1.07 ± 0.62

^1^ LOD = limit of detection.

**Table 5 animals-14-00889-t005:** Findings from the microbiological examination of alfalfa hay AH, *n* = 3) and meadow hay (MH, *n* = 3), expressed as colony-forming units (CFUs) per gramme of feed; reference range according to VDLUFA MB III 28.1.4 (2017) [21].

Group	AH (*n* = 3)		MH (*n* = 3)		Reference Ranges
	Min	Max	Min	Max	
MG 1 ^1^	1.1 × 10^7^	3.5 × 10^7^	6.0 × 10^6^	2.2 × 10^6^	≤3.0 × 10^7^
MG 2	5.8 × 10^5^	9.1 × 10^5^	<1.0 × 10^5^	2.0 × 10^5^	≤2.0 × 10^6^
MG 3	<1.0 × 10^5^		<1.0 × 10^5^	1.0 × 10^5^	≤1.5 × 10^5^
MG 4	2.0 × 10^5^	1.0 × 10^6^	6.3 × 10^3^	2.0 × 10^6^	≤2.0 × 10^5^
MG 5	<1.0 × 10^4^	6.4 × 10^4^	<1.0 × 10^4^	2.2 × 10^5^	≤1.0 × 10^5^
MG 6	<1.0 × 10^4^		<1.0 × 10^3^	<1.0 × 10^4^	≤5.0 × 10^3^
MG 7	1.0 × 10^4^	5.5 × 10^4^	8.2 × 10^3^	1.5 × 10^5^	≤1.5 × 10^3^
Clostridia	<100		<100	320	

^1^ MG = microbial group.

**Table 6 animals-14-00889-t006:** Findings of microbiological examination of alfalfa haylage (AS, *n* = 9), expressed as colony forming units (CFUs) per gramme of feed and reference range.

Group	AS (*n* = 9)		Reference Ranges (Grass Silage)
	Min	Max	
MG 1	<1.0 × 10^4^	7.3 × 10^6^	0.2 × 10^6^
MG 2	<1.0 × 10^4^	7.6 × 10^6^	0.2 × 10^6^
MG 3	<1.0 × 10^4^	<1.0 × 10^6^	0.01 × 10^6^
MG 4	<1.0 × 10^3^	1.8 × 10^4^	5.0 × 10^3^
MG 5	<1.0 × 10^3^	2.3 × 10^4^	5.0 × 10^3^
MG 6	<1.0 × 10^3^	<1.0 × 10^4^	5.0 × 10^3^
MG 7	<1.0 × 10^3^	1.0 × 10^4^	2 × 10^3^
Clostridia	<100		≤10^5^
Lactic acid bacteria	<1.0 × 10^4^	4.6 × 10^7^	n.s. ^1^

^1^ Not specified.

**Table 7 animals-14-00889-t007:** Daily feed intake of alfalfa hay (AH), alfalfa haylage (AS) and meadow hay (MH), expressed as % DM of BW and the median and [25th/75th] percentiles.

	AH (*n* = 11)	AS (*n* = 11)	MH (*n* = 11)	*p* Value
Daily feed intake (%)	2.10 [1.73/2.43]	2.15 [1.87/2.47]	2.12 [1.88/2.39]	0.194

**Table 8 animals-14-00889-t008:** Body weight on d0, d7 and d14 and weight difference between the measurement timepoints for feeding alfalfa hay (AH), alfalfa haylage (AS) and meadow hay (MH), expressed as the median and [25th/75th] percentiles.

Measurement Time	AH (*n* = 11)	AS (*n* = 11)	MH (*n* = 11)	*p* Value
d0	549 kg [519/570]	556 kg [526/579]	562 kg [529/574]	0.986
d7	556 kg [520/572]	557 kg [528/583]	558 kg [533/568]	0.967
d14	551 kg [522/574]	559 kg [526/575]	556 kg [531/572]	0.937
d7–d0	0.39% [−0.21/0.47]	0.29% [−0.32/0.79]	−0.29% [−1.00/0.68]	0.440
d14–d7	−0.53% [−0.90/0.40]	0.18% [−0.93/0.77]	−0.09% [−0.33/0.00]	0.761
d14–d0	−0.16% [−0.69/0.31]	−0.29% [−0.68/1.26]	−0.26% [−0.52/0.38]	0.982

**Table 9 animals-14-00889-t009:** Renal blood parameters and reference ranges on d0 and d21 of the experiment for feeding alfalfa hay (AH), alfalfa haylage (AS) and meadow hay (MH), expressed as the median and [25th/75th] percentiles.

Parameter	Time	AH (*n* = 11)	AS (*n* = 11)	MH (*n* = 11)	Reference Ranges [23]
Crea	d0	107	116	109	83.7–156.4
		[106/116]	[105/121]	[106/115]	
(mmol/L)	d21	91	87	104	
		[85/95]	[80/91]	[102/114]	
Urea	d0	4.20	4.29	4.60	2.51–7.34
		[4.09/4.62]	[4.21/4.61]	[4.10/4.80]	
(mmol/L)	d21	6.07	5.87	4.35	
		[5.78/6.28]	[5.57/6.16]	[4.09/4.84]	
Ca	d0	3.26	3.23	3.27	2.79–3.23
		[3.17/3.29]	[3.16/3.28]	[3.21/3.31]	
(mmol/L)	d21	3.27	3.26	3.19	
		[3.21/3.31]	[3.22/3.33]	[3.08/3.21]	
P	d0	1.07	1.04	1.05	0.66–1.50
		[1.01/1.13]	[0.99/1.12]	[1.00/1.16]	
(mmol/L)	d21	1.04	1.08	1.18	
		[1.01/1.12]	[0.93/1.11]	[1.11/1.22]	
Cl	d0	99.7	96.3	97.3	93.6–110.1
		[96.4/98.5]	[95.3/98.5]	[96.5/97.7]	
(mmol/L)	d21	93.2	95.6	96.9	
		[92.8/95.3]	[94.6/96.9]	[96.2/97.8]	
K	d0	4.06	4.04	4.36	2.60–4.65
		[3.68/4.49]	[3.63/4.32]	[4.04/4.55]	
(mmol/L)	d21	3.96	4.01	3.97	
		[3.82/4.03]	[3.93/4.13]	[3.80/4.08]	
Na	d0	135	135	134	137–144
		[133/137]	[134/137]	[134/136]	
(mmol/L)	d21	132	134	135	
		[131/135]	[133/135]	[134/136]	

**Table 10 animals-14-00889-t010:** Total protein, serum albumin, serum liver parameters and reference ranges on d0 and d21 of the experiment for feeding alfalfa hay (AH), alfalfa haylage (AS) and meadow hay (MH), expressed as the median and [25th/75th] percentiles.

Parameter	Time	AH (*n* = 11)	AS (*n* = 11)	MH (*n* = 11)	Reference Ranges [23]
Total protein	d0	63.8	64.5	64.6	57.8–78.7
		[62.1/64.8]	[62.4/66.6]	[63.3/66.0]	
(g/L)	d21	63.5	64.2	62.2	
		[62.8/64.3]	[63.1/65.1]	[60.5/63.6]	
Albumin	d0	34.5	33.8	34.1	27.3–37.0
		[33.9/35.0]	[33.3/34.7]	[33.3/35.3]	
(g/L)	d21	33.6	33.8	33.7	
		[33.3/33.9]	[33.4/34.6]	[33.1/34.4]	
AST ^1^	d0	336	329	343	213–627
		[319/399]	[313/364]	[312/362]	
(U/L)	d21	277	296	327	
		[271/309]	[281/321]	[303/378]	
GGT ^2^	d0	25.3	26.5	25.9	6.39–44.8
		[18.8/33.1]	[19.5/48.1]	[19.6/53.1]	
(U/L)	d21	21	17.9	22.8	
		[14.1/26.4]	[16/29.8]	[17.1/31.6]	
GLDH ^3^	d0	3.60	3.80	4.70	1.39–11.4
		[3.40/7.45]	[3.40/5.15]	[3.55/8.95]	
(U/L)	d21	3.10	3.10	3.40	
		[2.70/3.55]	[2.55/3.30]	[2.65/5.90]	
Bile acids	d0	4.80	5.50	6.20	<12
		[4.65/6.05]	[4.10/6.20]	[4.10/6.85]	
(µmol/L)	d21	4.00	5.00	4.40	
		[3.85/5.35]	[3.85/6.00]	[2.65/5.90]	
Bilirubin	d0	20.8	20.3	19.3	15.1–47.0
		[19.8/25.3]	[17.8/27.1]	[16.2/26.5]	
(µmol/L)	d21	19.1	18.3	20.8	
		[18.0/26.1]	[16.7/24.8]	[18.7/27.8]	

^1^ AST: aspartate aminotransferase, ^2^ GGT: gamma-glutamyl transferase, ^3^ GLDH: glutamate dehydrogenase.

**Table 11 animals-14-00889-t011:** Fractional excretion (FE) of electrolytes at different time points for feeding alfalfa hay (AH), alfalfa haylage (AS) and meadow hay (MH), data expressed as single values.

Parameter (%)	Time	AH (*n* = 6)	AS (*n* = 5)	MH (*n* = 4)	Reference Ranges (%)
FE_Ca_	Day 0–10	12.7/21.6	8.13/17.4/20.3	10.7/53.2	−0.16–6.72 [24]
	Day 11–21	6.48/11.3/20.8/24.8	15.8/22.0	6.69/41.3	
FE_P_	Day 0–10	0.00/0.08	0.00/0.01/0.02	0.00/0.01	0.04–0.16 [25]
	Day 11–21	0.00/0.01/0.02/0.04	0.00/0.00	0.00/0.00	
FE_Cl_	Day 0–10	1.00/2.03	1.20/1.67/1.77	1.58/1.69	0.48–1.64 [25]
	Day 11–21	0.39/0.71/0.96/1.66	0.45/1.19	0.69/1.27	
FE_K_	Day 0–10	79.1/148	77.0/115/142	68.5/149	23.9–75.1 [25]
	Day 11–21	58.5/88.6/101/127	51.5/87.0	77.0/125	
FE_Na_	Day 0–10	0.07/0.15	0.12/0.15/0.87	0.08/0.13	0.00–0.46 [25]
	Day 11–21	0.05/0.08/0.11/0.11	0.05/0.10	0.07/0.10	

**Table 12 animals-14-00889-t012:** Short chain fatty acids (SCFAs) in faeces in mmol/kg fresh matter (FM) for feeding alfalfa hay (AH), alfalfa haylage (AS) and meadow hay (MH) on d0 and d21, expressed as the median and [25th/75th] percentiles.

SCFAs (mmol/kg FM)	AH (*n* = 11)		AS (*n* = 11)		MH (*n* = 11)	
	d0	d21	d0	d21	d0	d21
Acetate	13.7	12.3	13.1	12.7	15.3	16.9
	[9.77/14.9]	[9.02/14.4]	[10.3/16.6]	[10.0/16.1]	[11.8/21.5]	[15.4/22.4]
Propionate	4.24	2.53	3.47	3.32	3.91	4.44
	[2.44/4.74] ^a^	[1.92/3.34] ^ab^	[2.66/4.44] ^a^	[2.44/4.24] ^a^	[3.25/5.37] ^a^	[3.71/5.51] ^ac^
i-butyrate	0.32	0.48	0.35	0.40	0.36	0.45
	[0.29/0.39]	[0.28/0.52]	[0.29/0.42]	[0.35/0.52]	[0.28/0.51]	[0.43/0.49]
n-butyrate	0.59	0.57	0.57	0.60	0.58	0.81
	[0.52/0.69]	[0.48/0.73]	[0.50/0.72]	[0.57/0.66]	[0.49/0.95]	[0.65/1.04]
i-valerate	0.24	0.3	0.27	0.36	0.31	0.36
	[0.22/0.31]	[0.23/0.39]	[0.22/0.47]	[0.24/0.42]	[0.21/0.41]	[0.32/0.41]
n-valerate	0.13	0.16	0.14	0.17	0.15	0.2
	[0.12/0.15]	[0.11/0.17]	[0.11/0.18]	[0.14/0.19]	[0.12/0.25]	[0.16/0.22]

^abc^ Different superscript letters indicate significant differences within a row according to the respective timepoints (*p* < 0.05).

## Data Availability

The data presented in this study are available on request from the corresponding author.

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
