# Peer review of "Nutrient Composition and Feed Hygiene of Alfalfa, Comparison of Feed Intake and Selected Metabolic Parameters in Horses Fed Alfalfa Haylage, Alfalfa Hay or Meadow Hay"

_animals, 2024, doi:10.3390/ani14060889_

Round 1

Reviewer 1 Report

Comments and Suggestions for Authors

General comments

This manuscript describes a feeding trial in which three types of conserved forages were used in the diet of three-year-old geldings. The manuscript is well written and has potential to make a useful contribution as a preliminary study.

However, there are some issues that need to be reviewed.

Detailed comments

Title:

Considering the comments made throughout the review, perhaps the authors should change the term “silage” to “haylage”.

Simple Summary:

As it is a summary of the work, the same comment (made previously) may apply.

Abstract:

The abstract must be changed according to the comments made throughout the manuscript.

Introduction:

3rd paragraph – In the characteristics of alfalfa, authors should also include that it is a crop that can be sensitive to acidic soils.

4th paragraph – When authors mention wrapped forage, the definition of haylage must also be introduced, in addition to silage (e.g. considering the references 18 and 38).

Materials and methods:

Study design

The authors refer to a 2-week wash-out period. It should be stated if this wash-out period was done between treatments. It is also not mentioned whether an adaptation period was carried out at the beginning of each treatment or whether this was included in the wash-out period.  

Diet

The composition of the commercial mineral supplement should be indicated, especially if it contains some quantities of calcium, phosphorus and magnesium.

Body weight

Why were the horses not weighed at the end of each feeding trial (at d21)?

Results:

Crude Nutrients

These results should be presented as a forage characterization. Therefore, results of Table 3 should be presented, like Table 4, as the mean ± SD. The way in which individual values are presented in two types of forage, and medians in the other, does not make sense, particularly the comparisons for differences. From the values published in several works, it is known that the crude protein and calcium content of alfalfa hay are normally higher than that of meadow hay.

In the present work, the authors report that the CP level of alfalfa hay (131.3±4.0) does not differ from that of meadow hay (82.9±4.4) (?) And for calcium (10.1±0.3 vs. 4.4±0.5) the same?

Given the dry matter value found for AS, from this point on, shouldn't AS be treated as alfalfa haylage?

Faecal DM

Strictly speaking, the first sentence should be similar to the first sentence in point 3.8. because on d0 all horses were still on MH.

Faecal SCFA

Superscript letters should be corrected. When there are no differences, it is not necessary to assign letters. For propionate values in d21, the letters a for AH, ab for AS and b for MH can be used.

Discussion

In a general way, authors should consider and discuss the designation of “alfalfa silage”, taking into account the average dry matter values.

7th paragraph:

…seems to be decisive…

…Silage bales with a dry matter content ≤ 55% … Is this content correct?

8th paragraph:

…0.51% and 0.02% of lactate and acetate… According to Table 4, aren't these values 0.01% and 0.013%?

On 14th paragraph when dry matter intake was discussed, authors should include some comments about the possible ingestion of some bed straw.

19th paragraph:

Authors should consider in the discussion other important regulatory mechanisms of calcium homeostasis (e.g. parathyroid hormones and calcitonin).

Also the Ca/P ratio should be included in the discussion.

28th paragraph:

Move the pH value.

Müller [39] reported that horses fed grass silage had a significantly higher faecal pH (6.78) than horses fed grass hay (6.64).

Conclusions:

Suggestion:

Instead of “highly resistant to climatic challenges” perhaps the authors could refer that … this type of forage can be an alternative forage in equine nutrition, in the context of climate changes.

Also, one of the concerns of using alfalfa as a single forage over prolonged periods in horse diets (whether hay or wrapped) is the high protein content. Mainly for horses with low or medium protein requirements. This issue should be included in the discussion.

Author Response

Dear Reviewer

We would like to thank you for a careful and thorough reading of the manuscript as well as the comments. We revised the manuscript accordingly (see File).

Best regards!

Reviewer 2 Report

Comments and Suggestions for Authors

Dear Authors,

I thoroughly enjoyed reading your paper, which presents valuable insights. However, I recommend adhering more closely to the journal's formatting guidelines. Specifically, the absence of line numbers in your manuscript complicates the review process, making it challenging to refer to specific sections for detailed feedback. Consequently, my suggestions for revisions will be somewhat broad in nature

Introduction:

In my opinion the introduction could benefit from a more detailed review of previous studies comparing alfalfa silage to other forages in equine diets. Specifically, highlighting gaps in current research that this study aims to fill would strengthen the rationale for the study. You should add a paragraph summarizing the findings of previous studies on alfalfa silage in equine diets and explicitly state how this study contributes new knowledge to the field.

To further underscore the significance of innovative approaches in equine nutrition education, the study DOI: 10.1016/j.jevs.2023.104537 provides crucial insights. It highlights the importance of teaching methodologies in enhancing veterinary students' engagement with equine nutrition. This perspective is vital for understanding how educational strategies can influence the adoption of forage alternatives in equine diets, making a strong case for the relevance of educational innovation alongside nutritional research

Given the significance of meadow hay as a traditional forage source for horses, it is essential to highlight the specific challenges it faces, particularly in the context of environmental variability and management practices. Factors such as climate change-induced droughts, irregular precipitation patterns, soil fertility variations, and harvesting/storage methods can significantly impact the availability and quality of meadow hay. Providing examples of regions or situations where these challenges are particularly acute would strengthen the discussion and illustrate the broader implications for equine nutrition. See: 10.1016/j.jevs.2022.103940.

To enhance the statement regarding the utilization of enzymatic digestion products and bacterial fermentation products by horses as sources of metabolized energy, particularly in the context of excessive starchy diets, additional references can be incorporated. Below is a suggested revision with added references:

"Horses utilize the products of enzymatic digestion, such as starch, in the small intestine as well as bacterial fermentation products, such as short-chain fatty acids (SCFAs), in the large intestine as sources of metabolized energy. Excessive intake of starch-rich diets can pose significant challenges to equine health, leading to various digestive disorders including colic, laminitis, and hindgut acidosis (10.1186/s12917-022-03289-2). High-starch diets can disrupt the delicate microbial ecosystem in the hindgut, causing dysbiosis and increased production of lactic acid, which can lower hindgut pH and predispose horses to colic and laminitis (10.3390/ani13061107). Furthermore, rapid fermentation of starch in the hindgut can result in the overproduction of SCFAs, leading to metabolic disturbances and suboptimal energy utilization (10.3390/ani12141740; 10.3390/ani10081334). Therefore, maintaining a balanced dietary starch content is crucial for optimizing equine digestive health and performance."

Methodology:

The description of the statistical analysis could be enhanced by explaining the choice of statistical tests and including a justification for the sample size. Therefore, you should include a subsection on statistical power analysis or sample size justification, detailing how the numbers were determined to be sufficient for detecting significant differences.

Results:

 Results are presented clearly with tables and figures that are easy to interpret.

Discussion:

The discussion integrates the results with existing literature, providing a comprehensive overview of how the findings fit into the broader context. However, the discussion could delve deeper into the implications of these findings for practical equine nutrition management and potential future research directions. I highly recommend you to expand the discussion on practical applications of feeding alfalfa silage to horses, including any considerations for transitioning horses to this feed source. Also, suggest specific areas for future research based on the study's limitations.

Conclusion:

 The conclusion succinctly summarizes the main findings and their importance.

Incorporating future perspectives, the authors can consider the following brief statement:

"In light of advancing digital communication channels, future perspectives include leveraging social networks and digital platforms to disseminate our research findings to a wider audience. Through engaging social media campaigns, collaboration with online communities, and interactive webinars, we aim to enhance awareness and understanding of alternative forage options in equine nutrition among stakeholders and enthusiasts." See: 10.3390/ani13223503.

Author Response

(The authors gave the same response as above.)

Reviewer 3 Report

Comments and Suggestions for Authors

The manuscript by Marlene Köninger et al. highlights the potential of alfalfa silage as an alternative forage option for horses. The study addresses the increasing need for forage alternatives due to periods of drought and the resulting shortage of traditional forage types. The main objective of the study is to assess the nutrient composition, feed hygiene, and the impact of alfalfa silage on feed intake, blood parameters, urine, and fecal parameters of horses when compared with alfalfa hay and meadow hay. Overall, the subject itself is surely worthy of investigation. However, some areas require improvement and clarification. Detailed comments and suggestions are provided below.

1- The title does not represent the experimental design and the objective of the study accurately. Please rewrite

2-  The simple summary section should be revised to enhance the clarity regarding the significance of the research topic, provide essential background information on cracker residue, explain the rationale for using alfalfa silage compared to alfalfa hay and meadow hay, and highlight the key and noteworthy results obtained from the study.

3-   hIncluding the objective(s) at the beginning of the abstract. The conclusion is too short and not reprsente your findings.

4-      Clarify the hypothesis of the study at the end of the Introduction section.

5-      There is a discrepancy in the numbering of cited references throughout the manuscript. The sequence begins with reference 38, then reference 18. Please revise the journal instructions for citing references. I think it is important to maintain a sequential numbering of references, starting from one.

6-  In section 2.2.; “….from November 2021 until April 2022” Did you mean April 2023?

7- How were 11 geldings randomly allocated to the experimental groups? how long the experiment lasted?

8-      In section 2.3; “…….the  horses had access to a pasture” add more description about the pasture.

9-      In section 2.3;

-          Blood samples were collected  ….. between 1 pm and 2 pm” What time was it after the morning meal? 3 or 4 hours?

-          How were urine samples collected? and how were they collected from 15 animals even though the number was 11?

10-  In section 2.5.2; why the number of samples is different?

11-  It is important to provide a detailed explanation of the procedure used for alfalfa silage production.

12-  In section 2.6.3; fecal sampling methods is missing especially for microbiological examination.

13-  The methods used for the fermentation parameters of AS were not mentioned in the M&Ms section. Please add.

14-  In all tables, it is recommended to provide the exact p-values instead of using "NS" or "*". Describe also the experimental groups and abbreviations in the tables' footnotes.

15-  Tables 9-11; specify the treatment, time, and interaction effects.

16- throughout the manuscript old references have been cited, please update.

Comments on the Quality of English Language

-

Author Response

(The authors gave the same response as above.)

Reviewer 4 Report

Comments and Suggestions for Authors

Dear authors,

A comprehensive study with relevant results to the equine veterinary field. Thank you.

Please consider the attached comments and re-check accuracy of all references.

Comments on the Quality of English Language

Dear authors,

the manuscript contains some English language spelling errors. Minor editing is recommended.

Author Response

(The authors gave the same response as above.)

Round 2

Reviewer 1 Report

Comments and Suggestions for Authors

Congratulations! The authors have made a good effort to improve the manuscript.

Just a small remark:

Faecal SCFA results:

For propionate, on d0 there is no need for superscript letters, as there are no differences. On d21, the superscript letters for AH should be a, for AS should be ab and for MH should be bc , because the value for AH is equal to AS, but different to MH. The value for AS is equal to AH and to MH.

Reviewer 3 Report

Comments and Suggestions for Authors

The authors adequately responded to all comments and performed all required modifications.

Comments on the Quality of English Language

-